

# The 4.2 cal ka BP Event in Northeastern China: A Geospatial Perspective

Louis A. Scuderi[1], Xiaoping Yang[2], Samantha E. Ascoli[1], and Hongwei Li[2]

[1]Department of Earth and Planetary Sciences, University of New Mexico, Albuquerque, NM, USA
[2]School of Earth Sciences, Zhejiang University, Hangzhou, Zhejiang Province, P. R. China

**Correspondence:** Louis A. Scuderi (tree@unm.edu)

**Abstract.** The Hunshandake Sandy Lands of northeastern China, currently a semi-arid lightly-vegetated region, were characterized by perennial lakes and forest stands in the early and middle Holocene. Well-developed dark grassland-type paleosols (mollisols) at the southern edge of the Hunshandake, OSL-dated to between 6.93 ± 0.61 and 4.27 ± 0.38 cal ka BP along with lacustrine sands in higher elevations that date to between 5.7 ± 0.3 and 5.2 ± 0.2 cal ka BP, and thick gray lacustrine sediments suggest a wetter climate. Between 4.2 and 3.8 cal ka BP the region experienced extreme drying that was exacerbated by headwater cutting of the overflow drainage that depleted the groundwater table through sapping. The region supported a robust population, the Hongshan Culture. The region depopulated post-4.2 cal ka BP and its people migrated out of the region, likely to the Yellow River Valley where they introduced their characteristic cultural elements to early Chinese civilization. Evidence for extreme and sudden environmental change in northeastern China, at and following the 4.2 cal ka BP event and like that we document in the Hunshandake, is widespread. However, no comprehensive overview exists. Here we discuss the relevant events in northeastern China and capture them in a spatially explicit Geographic Information Systems database that can be used to analyze the timing and spatial pattern of climate associated with the 4.2 cal ka BP event. This approach could serve as a prototype for a Global 4.2 cal ka BP event database.

*Copyright statement.* TEXT

# 1 Introduction

The Hunshandake Sandy Lands of northeastern China (Fig. 1), a region currently characterized by grasslands in its southern and eastern portion that overlie semi-stabilized aeolian deposits, and by aeolian sand sheets and dunes in its western portion. This extensive, desert-like landscape was covered by lakes and forests during the early and middle Holocene (Jiang et al., 2006; Yang and Scuderi, 2010; Yang et al., 2015). These bioclimatic regimes reflect significantly wetter conditions associated with an intensified monsoon precipitation up to 50% higher than current conditions (Yang et al, 2011, 2013). The region supported a significant population (Wagner, et al., 2013) that lived in small communities and relied on fishing and hunting (Liu and Feng, 2012; Wagner, et al., 2013). A sudden shift from wet to dry conditions in the Hunshandake, and for most of northeastern China,



occurs at ∼ 4.2 cal ka BP (Yang et al., 2015). While the primary driver appears to be linked to global scale change occurring at that time, in the Hunshandake it was exacerbated by rapid groundwater drawdown resulting from drainage capture. This combined climatic and hydrologic reorganization led to a rapid loss of surface water and a shift from green to sandy conditions over a few hundred years. This environmental shift produced regional depopulation with significant abandonment of sites across

the region until 3.6 cal ka BP (Liu and Feng, 2012; Wagner, et al., 2013; Yang et al., 2015). The change in environmental conditions in the Hunshandake is not unique within northeastern China. Evidence for a putative 4.2 cal ka BP event appears at sites across the deserts of China (Feng et al., 2006) and in many records from northeast China (Hong et al., 2001; Liu et al., 2002, 2010). Globally, the underlying cause for this event is controversial (Weiss and Bradley, 2010; Weiss, 2016, 2017). Part of the problem in deciphering this event includes a lack of understanding of the drivers of the global climate system at ca. 4.2

cal ka BP, understanding the spatial and temporal coherence of this event and resolving the sensitivity of different ecologic, hydrologic and geomorphic systems to forcings of this magnitude. In this paper we review the environmental change that took place in the Hunshandake bracketing the 4.2 cal ka BP event and place it in context relative to other northeastern China records. To better understand the event's temporal and spatial characteristics we analyze the existent literature from northeastern China, develop a geospatially explicit Geographic Information System from this literature and use it to map evidence for the 4.2 cal ka

BP event. We conclude with a discussion of how such a data structure and analysis approach might be used to better understand the 4.2 cal ka BP event globally and we provide the dataset for evaluation and analysis as an online supplement.

## 1.1 Study Area

China is characterized by a broad swath of deserts that extend between 38$^o$ and 46$^o$N. This desert belt is roughly divided at the Helan Mountains into a western portion of "true" hyper-arid and arid deserts, and a slightly wetter eastern portion consisting

of lightly vegetated and stabilized semiarid to dry subhumid "Sandy Lands". The Hunshandake Sandy Lands (Fig. 1, elevation range 1100-1400m), along with the Horqin and Hulun Buir, are found on the eastern edge of this desert belt in northeastern China. Ecologically the Hunshandake is a semi-arid grassland ecosystem underlain primarily by aeolian sandy soils. Monthly temperatures range from -18.3 °C in January to +18.5 °C in July with a mean annual precipitation 150 to 450 mm, falling mainly during the summer months.

## 1.2 Holocene Climatic History

The Hunshandake Sandy Lands can be subdivided into southern, western and eastern units. The southern part of the region is characterized by low hills vegetated by a thin cover of grasses and shrubs and has an absence of standing water. The western part of the region is primarily low rolling topography with a grassland cover and standing lakes while the eastern portion is grassland and low rolling vegetated dunes with dry lake beds. While all three are currently semi-arid lightly vegetated

grasslands, they differed in their response to the 4.2 cal ka BP event.

The Holocene environment of this area is discussed in detail in Yang et al. (2015). We briefly summarize the findings below. Figure 2 illustrates two well-developed dark grassland-type paleosols (mollisols) at the southern edge of the Hunshandake that are identifiable and can be traced across the entire region. Lacustrine sands underlying these paleosols indicate an earlier





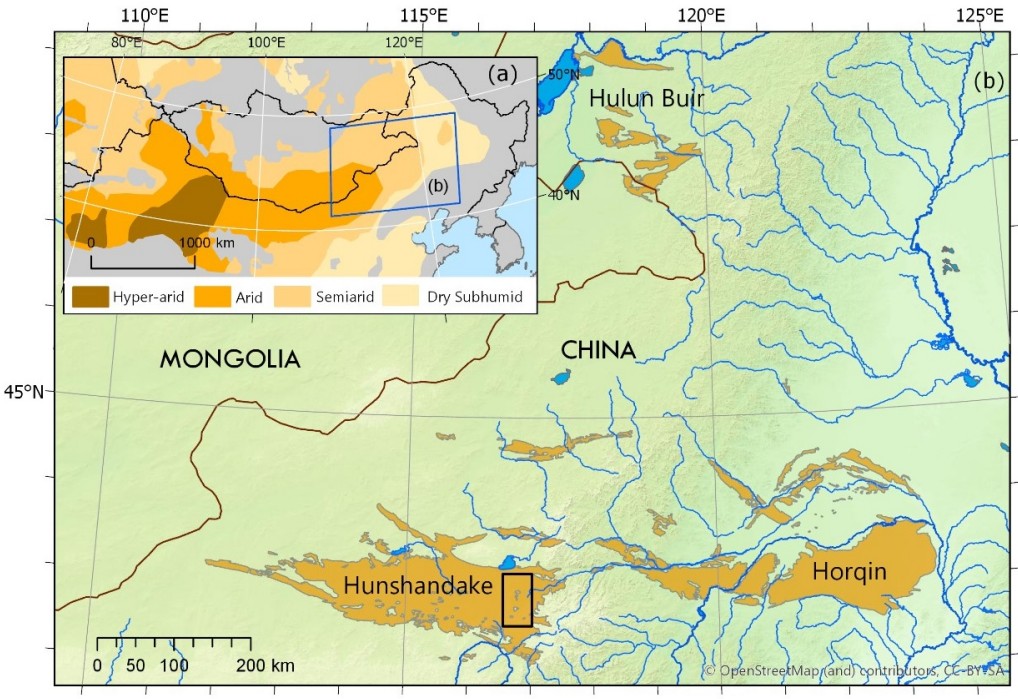

**Figure 1.** Sandy Lands of northeastern China. Inset a. Distribution of deserts across China by climate type. b. Sandy land distribution. Boxed area in the Hunshandake Sandy Lands is study area reported in this study.

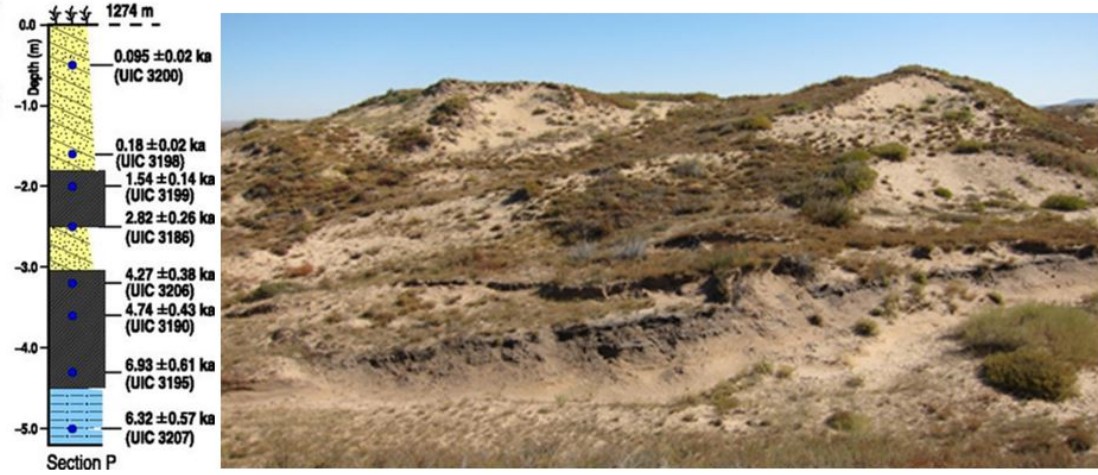

**Figure 2.** Southern Hunshandake. Left: Section P from Yang et al. (2015) showing paleosols and sandy units which overlie an earlier lacustrine unit. Right: Sampled exposure showing the two paleosols.



lake/wetland environment. The lower paleosol is OSL-dated to between 6.93 ± 0.61 and 4.27 ± 0.38 ka BP and suggests a period of wetter climate that rapidly transitioned to dry conditions at ca. 4.2 cal ka BP. The southern part of the Hunshandake, as indicated by the second paleosol dated to between 2.82 ± 0.26 and 1.54 ± 0.14 ka BP, returned to green conditions again at ca. 2.8 ka BP and maintained this state for 1,500 yrs. Dune sediments reflecting an active aeolian environment have dominated since 1.3 cal ka BP.

The eastern Hunshandake (Figs. 3 and 4) while exhibiting evidence of 4.2 cal ka BP event in terms of paleosols and shoreline features does not indicate a return to grassland conditions between 2.8 and 1.5 ka BP with sandy conditions persisting since the onset of the 4.2 cal ka BP event. As Yang et al. (2015) found this difference was associated with redirection of drainage from its northerly flow towards Dali Lake at ca. 4.5 cal ka BP by capture of the surface waters and groundwater table by the Xilamulun River. This may be reflected in the longest period of low stands of both Dali (Xiao et al., 2008) and Daihai (Xiao et al., 2004, 2006) lakes between ca. 4.5 and 3.8 cal ka BP. The eastward drainage shift resulting from this capture, coupled with channel entrenchment via groundwater sapping resulted in long-term drying of the eastern Hunshandake and abandonment of the region by the Hongshan culture (Peterson et al., 2010) ca 4.5-4.2 cal ka BP. Recent mapping suggests a lack of artifacts in the eastern Hunshandake between 4.3 and 3.5 cal ka BP (Liu and Feng, 2012; Wagner, et al., 2013). In the eastern Hunshandake Sandy Lands, Hongshan artifacts are found primarily within and below the 4.2 cal ka BP paleosols and shorelines while Bronze Age artifacts (Fig. 3b), which appear in the region ca 3.6 cal ka BP (Wagner et al., 2013), lie on or above the 4.2 cal ka BP paleosol.

## 2   Results

### 2.1   Deriving and Understanding the Regional Signal in Northeast China

The 4.2 cal ka BP event continues to puzzle scientists with respect to its spatial extent, triggering mechanisms, and regional to global characteristics. As shown above, the difference in response between two sites separated by less than 100 km illustrates some of the complications that can arise from point source reconstructions. In northeastern China records are dominated by evidence from sediments and pollen from lacustrine and aeolian environments. At times these records present conflicting views of the 4.2 cal ka BP event. In the following we discuss some of the issues with interpreting regional constructions from this diverse set of records and provide an approach to reconstructing the regional signal of this event in northeastern China.

Parmesan and Yohe (2003) note that detection of a climate change signal is a search for spatially and temporally coherent sign-switching patterns. This search is complicated by the time transgressive and rate response differing nature of complex interacting systems as well as possible spatially diffusive responses across a landscape. While records from individual sites provide snapshots of what occurred at a given locality, interpretation is often complicated by significant uncertainty in dating and possible regionalization of the observed response.

Placing the Hunshandake record in context relative to northeast China, and within the larger global context, requires the assembly of records from many diverse sources into a single data repository/database with well-defined and consistent termi-





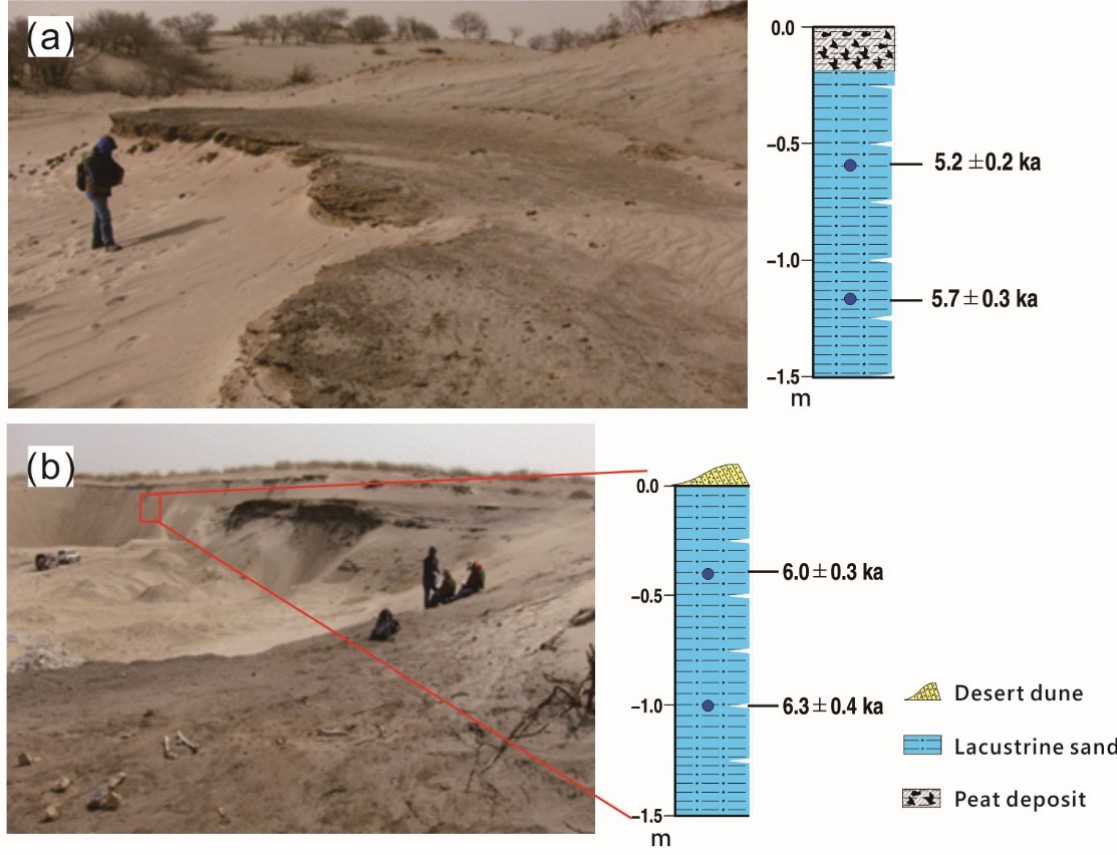

**Figure 3.** Eastern Hunshandake lacustrine deposits capped by paleosols. a. Eastern Hunshandake Paleosol/lakeshore (Section I from Yang et al., 2015) b. Bronze age burial (lower left) on the surface of the lake shore exposed by deflation (Section E from Yang et al., 2015).

nology. Without such efforts such "databases" are simply collections of facts and observations rather than knowledge generation tools.

To "redefine" the problem and to minimize database problems associated with definitional confusion (semantic) and locational (spatial) and dating (temporal) uncertainty, we approached the 4.2 cal ka BP event from a different perspective, namely by letting the body of existing data define the parameters of the information space we were exploring and database structure. To do so we utilized conceptual spaces for defining the informational architecture of our reconstruction of the 4.2 cal ka BP event in northeast China. We note that such information context extraction may be useful in the identification of similar and differing responses between regions and understanding how these expressions of the 4.2 cal ka BP event may have varied spatially and temporally.





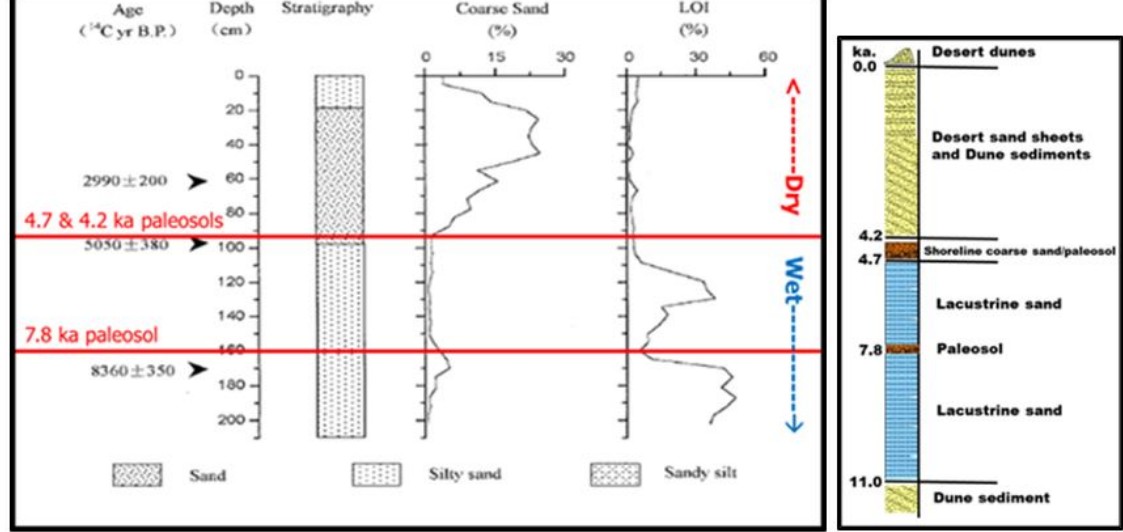

**Figure 4.** Left: Core records from Xiaoniuchang (after Liu et al, 2002). Right: General Holocene reconstruction of eastern Hunshandake Sandy Lands (Yang et al. 2015). The coarse sand fraction increases significantly following the 4.2 cal ka BP event with sediments dominated by desert sand sheets and dunes.

## 2.2 Database Creation

We collected and analyzed 60 peer reviewed articles (listed 1-60 in the Supplementary material references) pertaining to middle-to-late Holocene climate and environmental conditions in northeast China. This analysis included text, figures and tables. Using a web enabled open access semantic parser (Word Count Tool: High Star App, 2017: https://wordcounttools.com/)

5 we extracted ∼430,000 words (excluding simple words- a, it, the, etc. as well as dates). While the word counter provides extensive information about the word count structure, in our analysis we used only raw word count statistics. From this output we identified the 30 most used words from each article (224 critical words distributed across the 60 articles). These critical words appear 28,289 times (∼6.5%) in the articles analyzed.

  We organized the words by similarity and number of occurrences (Fig. 5) and derived database tables (Fig. 6) with explicit

10 geocoding that were input into ARCInfo (ESRI, 2018) for geospatial analysis. We also included paleoclimate reconstructions of Mid-Holocene (6.0 ka BP) climatic parameters derived from Coupled Model Intercomparison Project 5 (CMIP5) data. This downscaled data (30-second resolution) was calibrated (bias corrected) using WorldClim 1.4 (Hijmans et al., 2005) as a baseline for 'current' climate and was used as a reference for conditions prior to 4.2 cal ka BP. An example showing modelled 6.0 ka BP June precipitation (Fig. 7) is useful for illustrating the wide range of a single climatic variable across our northeast

15 China study area ( 1500 km2) and the potential difficulties arising from using individual site (point) data in assessing the impact of the 4.2 cal ka BP event.





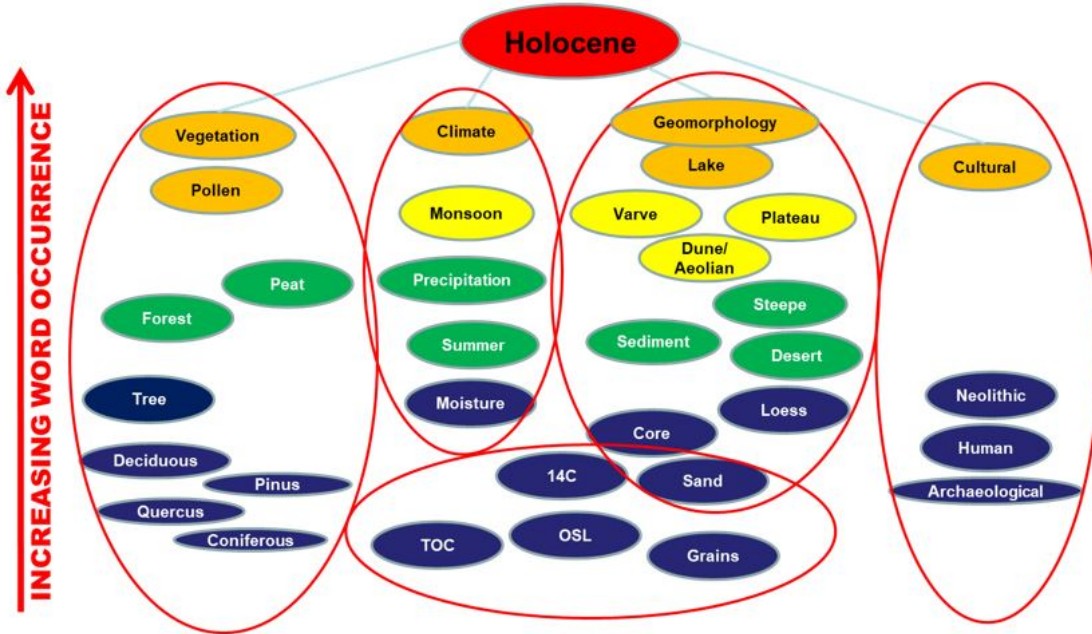

**Figure 5.** Topic Map. Word extraction organized by number of words (Left axis: Red highest, Blue lowest) and by four major Topic Groups (Bottom Axis: Left- Vegetation, Left Middle- Climate, Right Middle- Geomorphic/Sedimentologic Type, Right – Cultural). Lower Center- Analysis Type.

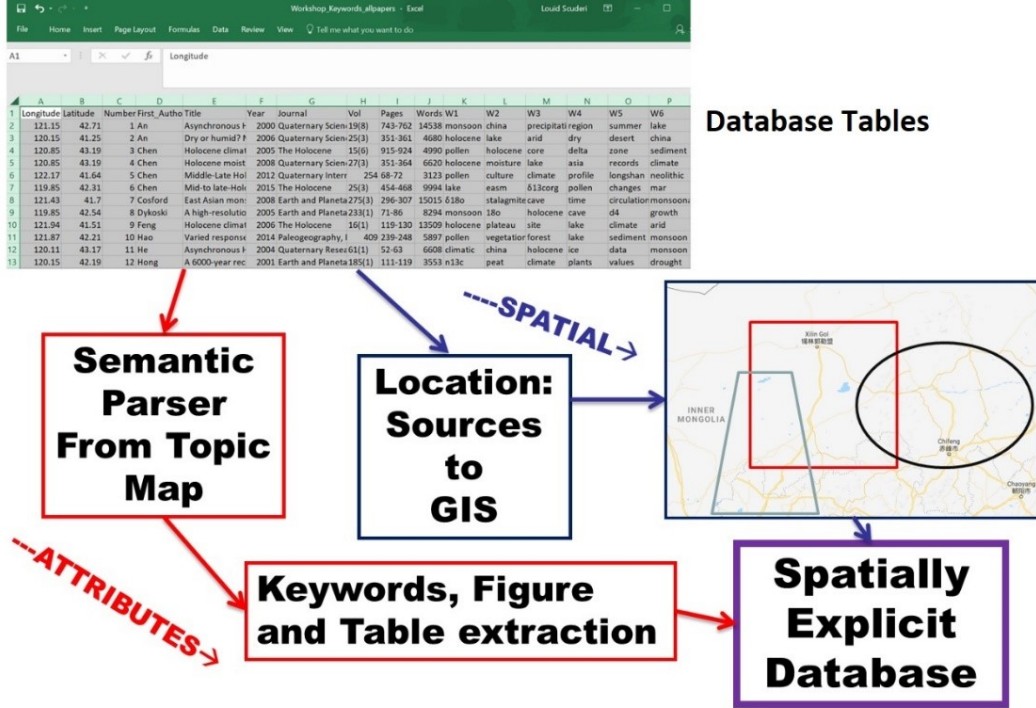

**Figure 6.** Overview of the processing schema for database production. Each article was geocoded and combined with tables derived from keywords to produce the final GIS compatible spatially explicit database for analysis.




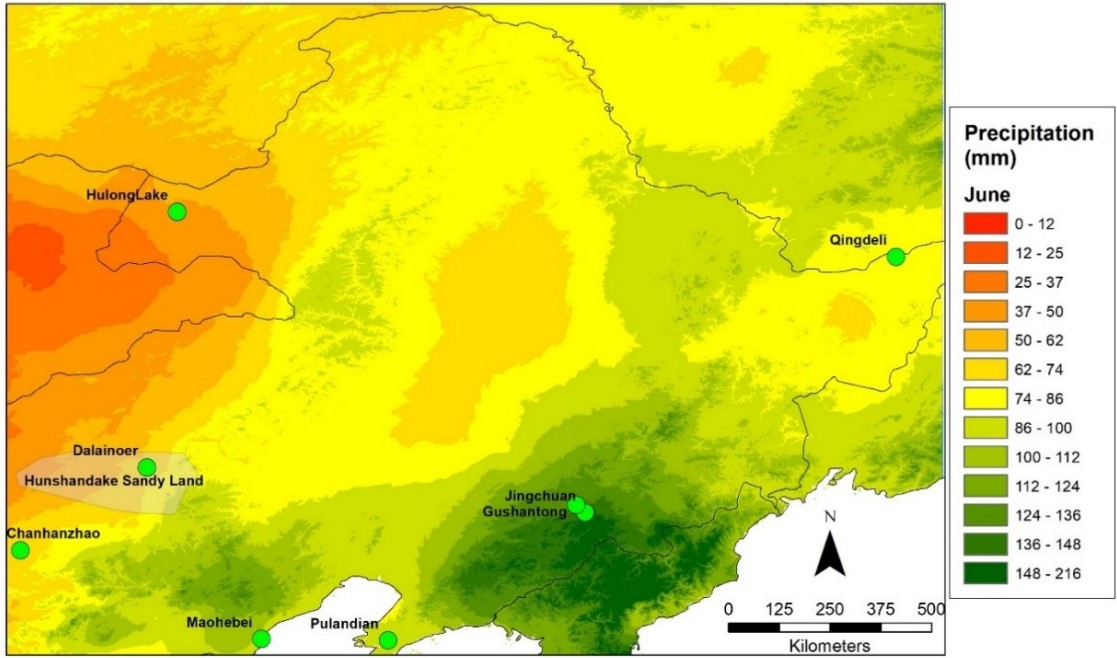

**Figure 7.** Modelled June precipitation (mm) for 6.0 ka BP (Hijmans, et al. 2005). Eight sites derived from An et al. (2000) are shown (green dots) to illustrate the incomplete spatial distribution of records within individual research records across the region.

Finally, following the lead of Parmesan and Yohe (2003), we identified the 4.2 cal ka BP state change (yes, no) and rate and direction of change (rates: +2 to -2 with 0 being no change, direction: positive or negative indicating an increase or decrease in the measured variable) from the text and figures in each article and linked them in the database to the specific topic group they were derived from. This allows us to analyze and map the distribution, intensity and directionality of 4.2 cal ka BP marker
5    events, or lack thereof, of the different types of evidence available.

## 3   Analysis

An example of regional results for change at 4.2 cal ka BP (presence or absence of 4.2 cal ka BP event) is illustrated in Fig. 8. It is clear from the distribution of points that the signature of a 4.2 cal ka BP event is strong but not universal at all sites across northeast China with 23% of sites (18/77) reported in the 60 papers showing inconclusive or no evidence of the 4.2 cal
10   ka BP event. Spatial analysis of the distribution of reveals an interesting pattern with coastal and sites over 750km from the





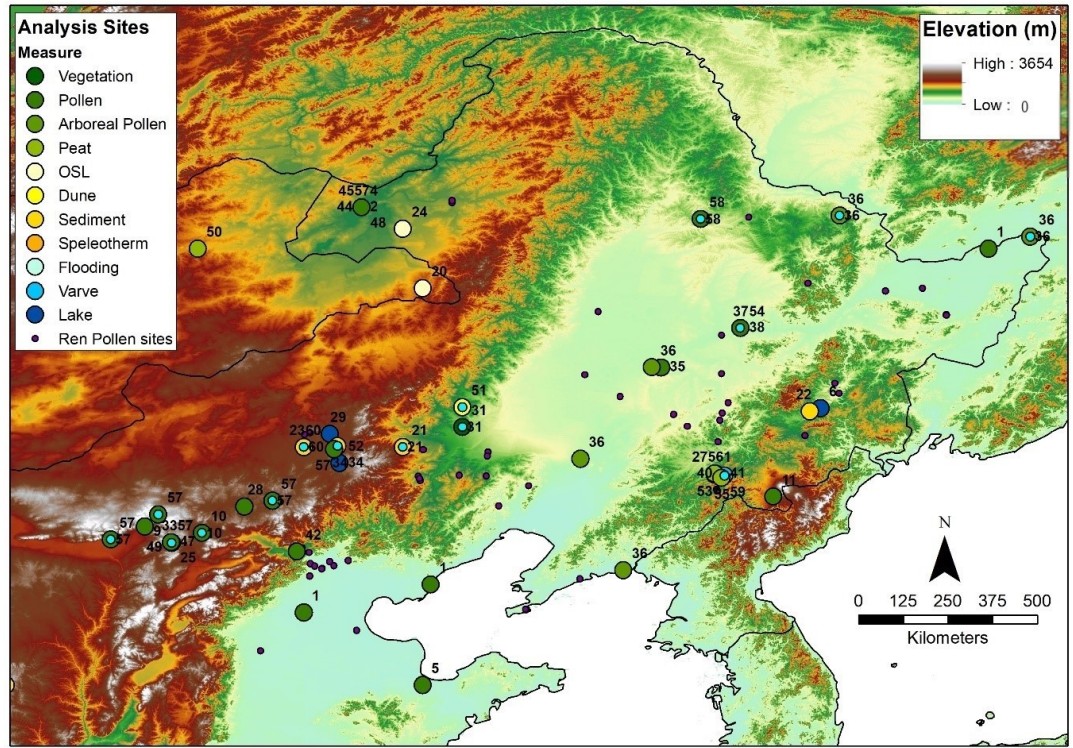

**Figure 8.** 4.2 cal ka BP event analysis sites by type of evidenced measured (numbered site information in Appendix 1). Note the lack of sites in the northern portion of the region (Heilongjiang Province and NE inner Mongolia). Light blue dots within individual measures indicate sites with no change across the 4.2 cal ka BP event Small purple dots are pollen sites from Ren and Zhang (1998) derived from published work and unpublished dissertations. While not all sites cover the 4.2 cal ka BP event we include the Ren and Zhang data to show the distribution of late-Holocene paleoclimatic reconstruction work in NE China.

coast showing consistent evidence for change at 4.2 cal ka BP. However, an intermediate band of higher elevation sites between 300 and 500km from the coast that are primarily derived from measures of vegetation type or abundance shows inconsistent evidence for the 4.2 cal ka BP event. Whether this is the result of interpretation errors, lack of sensitivity and/or temporal resolution, or in fact the actual absence of environmental markers for the event will require additional research.

5     While there appears to be a significant level of coherence that suggests a relatively strong 4.2 cal ka BP signature across northeast China there are examples of local results that are inconsistent with this generalization. As we report in this paper and in Yang et al. (2015) within the Hunshandake sites separated by as little as 100km (Haoluku 42.95N, 116.76E, Xiaoniuchang 42.62N, 116.82E) exhibit signals that differ significantly. Liu, et al. (2002) suggested, that these differences might be due to a combination of elevation (Xiaoniuchang 1460m, Haoluku 1295m), local conditions (Xiaoniuchang at the edge of a lake and



Haoluku at its center) and transport of sediment due to changing environmental conditions (Xiaoniuchang dried up earlier than Haoluku which was still experiencing inflow of coarse sediments). Dating precision and uncertainty may also result in apparent differences between local sequences both within the Hunshandake and in other localities that we document. While we currently do not have the ability to recognize different parts of the 4.2 cal ka BP event in northeast China, as has been done for the central and western Mediterranean region (Magny et al., 2009), it is clear from the records we analyzed that the event is likely to have had multiple phases in northeast China.

## 4 Conclusions

Lack of integration of data into a scientifically credible, globally assembled, information platform with consistent terminology and definitions to guide scientific enquiry hinders understanding of the 4.2 cal ka BP event. Creation of such an information platform allows researchers to ask questions about the spatial distribution and environmental indicators that characterize the 4.2 cal ka BP event. Such a platform can expand the range of research questions that can be tackled, facilitate innovative and collaborative research, allow data sharing and comparison of results and to facilitate development of innovative analytic tools. As Yin (2005) noted, the end goal of this type of research "is a data structure that sits firmly upon the deep-seated, some might say, hard-wired, natural structures of the information architecture".

Using this approach, we showed the presence of a strong and coherent signal for the 4.2 cal ka BP event in northeastern China, albeit with local and regional variation that complicates interpretation. Much of this "complication" may be the result of the use of different standards, differing interpretations of the data, data gaps, and differential spatial and temporal responses of indicators analyzed and reported. While much work remains, our prototype database approach, guided by sematic analysis of the literature and georeferencing of existent data sources, can serve as a guide to the assembly of a larger scale global 4.2 cal ka BP database that will allow better understanding of this climatic event. The reader is encouraged to use the dataset found in the online supplement to both explore 4.2 cal ka BP relationships in northeast China and to possibly guide the development of similar databases for other parts of the world.

## Appendix A

**A1**

*Author contributions.* LS and SA developed the Geographic Information System database and LS performed the model runs. LS, and HL prepared graphics. LS prepared the manuscript with contributions from all co-authors.

*Competing interests.* No competing interests

*Acknowledgements.* The authors would like to thank the 4.2 cal ka BP Conference organizers for providing a venue for this work. LS and SA would like to thank the University of New Mexico for travel and computing support. XP and HL would like to thank the School of Earth Sciences, Zhejiang University, Hangzhou, Zhejiang Province, P. R. China for continued support. SA would like to thank the Leonard Foundation for research support.

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
