# Peer review of "The 4.2 cal ka BP Event in Northeastern China: A Geospatial Perspective"

_Climate of the Past, 2018_

## Referee Comment (RC1) · Anonymous Referee #1 · 10 Nov 2018

General comments:

The spatial pattern of the 4.2 ka BP climatic event is a key point to understanding the mechanism responsible for operating the Earth's climate system on millennial to centennial scales. In northeastern China, there are three large Sandy Lands (currently vegetated and with semi-stabilized dunes) that experienced changes between deserts and forest grasslands in response to climate changes in the geological past. Previous studies on eolian and lacustrine records suggest that abrupt environmental changes occurred in these Sandy Lands during the Holocene. However, an integrated view of these events is still far beyond reach. In this manuscript, the authors developed a spatially explicit Geographic Information System to capture regional climatic events and analyze the timing and spatial pattern of the 4.2 ka BP event, not only revealing

the regional manifestation of the event but providing a promising approach for research of abrupt climate changes as well. I recommend acceptance of this manuscript for publication in CP after revisions.

Specific comments:

1. Page 2 Line 23. Delete "mean" because the precipitation is given as an amount range rather than a single amount.

2. Figure 1 on Page 3. Show more numbers of the latitude in panel for the readers to recognize latitudinal intervals easily.

3. Page 2 Line 30. Better to change "the 4.2 cal ka BP event" to "climate changes".

3. Page 4 Lines 8-13. I can understand the result of the eastward shift in the drainage of the Xilamulun River: water level lowering at Dali Lake and environmental drying in the eastern Hunshandake. But I have wondered whether the Horqin Sandy Land became somewhat wet in this condition in terms of its location at the lower reaches of the Xilamulun River.

4. Figure 7 on Page 8. Change "HulongLake" to "Hulun Lake"; "Dalainoer" to "Dali Lake"; "Jingchuan" to "Jinchuan"; and "Qingdeli" to "Qindeli".

5. Page 8 Line 10. Add some words after "the distribution of ".

6. Page 10 Lines 2-3. Additionally how do you consider the effects of the topographic relief and groundwater runoff in the study region?

7. A newly published paper (Xiao et al., 2018. The 4.2 ka BP event: multi-proxy records from a closed lake in the northern margin of the East Asian summer monsoon. Climate of the Past 14, 1417-1425.) refers to Hulun Lake shown in Figs 1 and 7 in this manuscript and may be of some help to this study.

Related aspects: 1. Does the paper address relevant scientific questions within the scope of CP? Yes. 2. Does the paper present novel concepts, ideas, tools, or data?

Yes. 3. Are substantial conclusions reached? Yes. 4. Are the scientific methods and assumptions valid and clearly outlined? Yes. 5. Are the results sufficient to support the interpretations and conclusions? Yes. 6. Is the description of experiments and calculations sufficiently complete and precise to allow their reproduction by fellow scientists (traceability of results)? Yes. 7. Do the authors give proper credit to related work and clearly indicate their own new/original contribution Yes. 8. Does the title clearly reflect the contents of the paper? Yes. 9. Does the abstract provide a concise and complete summary? Yes. 10. Is the overall presentation well structured and clear? Yes. 11. Is the language fluent and precise? Yes. 12. Are mathematical formulae, symbols, abbreviations, and units correctly defined and used? Yes. 13. Should any parts of the paper (text, formulae, figures, tables) be clarified, reduced, combined, or eliminated? No. 14. Are the number and quality of references appropriate? Yes. 15. Is the amount and quality of supplementary material appropriate? Yes.

---

## Referee Comment (RC2) · Anonymous Referee #2 · 11 Nov 2018

The paper by Scuderi et al. presents an example of geospatial analysis on the 4.2 events database by using the new conceptual spaces for defining the informational architecture of the reconstruction of this event in northeastern China. It's a new approach to integrate the climate event in a large region and even on the continental scale. However, there are still some issues need to be considered in the integration with this geospatial analysis. I recommend the acceptance of this manuscript for publication in the journal of Climate of the past after moderate reversions.

The description for the eastern, southern and northern units of Hunshandake Sandy Lands is rather unclear. Is it the yellow shaded region in figure 1, labelled with Hunshandake, was subdivided three units or the box region, namely the study area reported in this study, was subdivided three units? I suggest the authors to rewrite this

part and describe this clearly. The authors may use simple plots to indicate these three units.

How the CMIP5 data was generated from the coupled model intercomparison project 5 data? Is it the average of the multiple model outputs or just a single model output (which model) for 6 ka? Although the wide range of a single climate variable (e.g. June precipitation in the text) at the study site is somewhat useful to illustrate the difficulty using a single site in assessing the impacts of the 4.2 event, the changing direction and rates are more important to assessing the influence of the event. So, the varied climate changes observed in the study area in different years might be more robust to illustrate this difficulty. In addition, the paleoclimatic proxy usually reflect the mean climate condition and the relative changes, so using the absolute value of June precipitation to illustrate the complex of using single site in studying the climate change is some unsuitable. Maybe using the seasonal or annual climate condition changes is more appropriate.

In the 77 sites reported in 60 published papers, some sites might be reported several times, which may bias the evaluation inevitably. How to deal with these repeats in different publications should be considered in the geospatial analysis.

The 4.2 ka event is hardly to be extended to 3.0 ka and even later. A return to grass land condition between 2.8 and 1.5 ka BP at eastern Hunshandake can't be regarded as a different signal for 4.2 ka event. So, I suggest the authors should also double check the response of 4.2 ka event at other sites and place this event within a certain period, although the chronology uncertainty could be a factor broadening this period slightly.

---

## Author Comment (AC1) · 20 Dec 2018

Authors Response: The 4.2 cal ka BP Event in Northeastern China: A Geospatial Perspective First, the authors would like to thank the reviewers for their valuable and insightful comments which will materially improve both the paper and the database from which the analysis was derived. Below we outline the changes that we will make to the manuscript in response to both reviews. Referee comments in first. Responses follow. Author's changes to be made in the manuscript indicated last.

Please note that some comments for reviewers 1 & 2 are dealt with jointly so we have included both here.

Response to Reviewer 1's comments: 1. Page 2 Line 23. Delete "mean" because the

precipitation is given as an amount range rather than a single amount.

Agreed. "Mean" was not an appropriate usage.

Line 23 changed to: ". . . . . .in July with annual precipitation across the region between 150 to 450 mm . . . ."

2. Figure 1 on Page 3. Show more numbers of the latitude in panel for the readers to recognize latitudinal intervals easily.

Agreed that this will improve interpretation of figure 1.

Figure 1 modified to show a finer latitude graticule (see also comments to reviewer 2 regarding further modification of this figure to better delineate the three regions of the Hunshandake that we discuss in the manuscript).

3. Page 2 Line 30. Better to change "the 4.2 cal ka BP event" to "climate changes".

Agreed that this is a more appropriate wording.

Page 2 Line 30 changed to: ". . . . . . in their response to climate change at $\sim$4.2 cal Ka BP."

3. Page 4 Lines 8-13. I can understand the result of the eastward shift in the drainage of the Xilamulun River: water level lowering at Dali Lake and environmental drying in the eastern Hunshandake. But I have wondered whether the Horqin Sandy Land became somewhat wet in this condition in terms of its location at the lower reaches of the Xilamulun River.

We agree that the Horqin Sandy Land likely became somewhat wetter with the addition of drainage from the Hunshandake via the Xilamulun River post 4.2 Ka. We will now note this in the text as follows;

Page 4 Line 13. ". . . . . .resulted in long-term drying of the eastern Hunshandake, likely moisture enhancement of the downstream Horqin Sandy Land, and subsequent abannone

donment of the Hunshandake by the Hongshan culture (Peterson et al., 2010) ca. 4.5-4.2 cal ka BP."

*Additionally, we believe that this aspect of environmental change impacting the downstream Horqin Sandy Land is an important event in the region which requires further research.

4. Figure 7 on Page 8. Change "HulongLake" to "Hulun Lake"; "Dalainoer" to "Dali Lake"; "Jingchuan" to "Jinchuan"; and "Qingdeli" to "Qindeli".

We note that the original texts used in the research are often inconsistent in terms of naming places. This created problems with database creation. Our approach was to utilize the most commonly used name found in the literature. A future version of the database will need to include data columns with alternative names for each site.

Figure 7 has been changed to reflect these name changes as requested by the reviewer.

5. Page 8 Line 10. Add some words after "the distribution of ".

Page 8 Line 10. This somehow disappeared when we converted to the LaTeX format.

Text added as follows:

". . . Spatial analysis of the distribution of these records reveals an interesting pattern . . .. . .."

6. Page 10 Lines 2-3. Additionally, how do you consider the effects of the topographic relief and groundwater runoff in the study region?

We agree that this is a potentially important point and have modified the sentence and added text. The new text on Page 10 Lines 2-3 will read;

". . ... inflow of coarse sediments). Dating precision and uncertainty, as well as variability in existing groundwater conditions and local and regional differences in topographic

relief, may also result in differences. . . . . . ."

*We do note that capturing groundwater levels at the time of the 4.2 ka BP event is difficult and we know of no existing data that allows us to address this issue directly. As far as the issue of topographic relief impacting the response, we note that a DEM is included in the database and that users of the database can explore this possible driving mechanism.

7. A newly published paper (Xiao et al., 2018. The 4.2 ka BP event: multi-proxy records from a closed lake in the northern margin of the East Asian summer monsoon. Climate of the Past 14, 1417-1425.) refers to Hulun Lake shown in Figs 1 and 7 in this manuscript and may be of some help to this study.

We did not have access to this paper when we created and submitted the manuscript.

We have now included references to this paper in several places in our discussion. We will also add a record to the database to reflect the conclusions and interpretations in this paper. As well, the reference will be added to the reference list in the main portion of the manuscript.

Response to Reviewer 2's comments:

1. The description for the eastern, southern and northern units of Hunshandake Sandy Lands is rather unclear. Is it the yellow shaded region in figure 1, labelled with Hunshandake, was subdivided three units or the box region, namely the study area reported in this study, was subdivided three units? I suggest the authors to rewrite this part and describe this clearly. The authors may use simple plots to indicate these three units.

Agreed. This will make the discussion clearer.

We have redrafted the figure to now include a subfigure that outlines the three areas as well as the finer latitude delineation requested by reviewer 1.

2. How the CMIP5 data was generated from the coupled model Intercomparison Project 5 data?

First, for some background we paraphrase from the website http://www.worldclim.org/paleo-climate1 which we will include with the online documentation of our NE China database.

"WorldClim 1.4 downscaled paleo climate: The data available here downscaled climate data from simulations with Global Climate Models (GCMs). The original data was made available by (CMIP5). These data were downscaled and calibrated (bias corrected) using WorldClim 1.4 as baseline 'current' climate. The file format is GeoTIFF.

The data available were produced starting with the projected change in a weather variable. This is computed as the (absolute or relative) difference between the output of the GCM run for the baseline years ("pre-industrial" for past climate studies) relative to an earlier time interval. These changes were interpolated to a grid with a high ($\sim$ 1 km) resolution. The assumption made is that the change in climate is relatively stable over space (high spatial autocorrelation).

The results are then downscaled. Downscaling was accomplished using observed weather data to describe relationships between larger-scale climate variables (e.g. atmospheric pressure at 1000 m) and local surface climate variables (e.g. surface rainfall). This relationship was then applied to GCM output under the assumption that the GCMs perform best for the larger-scale variables; and that the relationships found remain valid in a changed climate."

Downscaled Mid-Holocene (6 ka) data is available from WorldClim for 9 models (BCC-CSM1-1, CCSM4, CNRM-CM5, HadGEM2-CC, HadGEM2-ES, IPSL-CM5A-LR, MIROC-ESM, MPI-ESM-P, MRI-CGCM3) at four spatial resolutions (grids of 10, 5, 2.5 minutes and 30 seconds).

A list of model data available follows;

Table Supplement 1: WorldClim 6 ka Reconstructions. 1. BCC-CSM1.1 Beijing Climate Center Climate System Model http://forecast.bcccsm.ncc-cma.net/web/channel-43.htm 2. CCSM4 Community Climate System Model (CCSM) http://www.cesm.ucar.edu/models/ccsm4.0/ 3. CNRM-CM5 National Centre for Meteorological Research https://portal.enes.org/models/earthsystem-models/cnrm-cerfacs/cnrm-cm5 4. HadGEM2-CC Hadley Global Environment Model 2 - Carbon Cycle https://view.es.doc.org/renderMethod=name&type=cim.1.software.ModelComponent&name=HadGEM2-CC&project=CMIP5 5. HadGEM2-ES Hadley Global Environment Model 2 - Earth System https://view.es.doc.org/renderMethod=name&type=cim.1.software.ModelComponent&name=HadGEM2-ES&project=CMIP5 6. IPSL-CM5A-LR Institut Pierre Simon Laplace Model http://www.glisaclimate.org/node/2218 7. MIROC-ESM Model for Interdisciplinary Research on Climate-Earth System Model http://www.glisaclimate.org/model-inventory/model-for-interdisciplinary-research-on-climate-miroc-version-32-medres 8. MPI-ESM-P Max‐Planck‐Institute Earth System Model https://www.mpimet.mpg.de/en/science/models/mpi-esm/ 9. MRI-CGCM3 Meteorological Research Institute Coupled Global Climate Model Version 3 http://www.glisaclimate.org/node/2542

Table Supplement 2: Bioclimatic Variables. Abbreviation Variable BIO1 Annual Mean Temperature BIO2 Mean Diurnal Range (Mean of monthly (max temp - min temp)) BIO3 Isothermality (BIO2/BIO7) (* 100) BIO4 Temperature Seasonality (standard deviation *100) BIO5 Max Temperature of Warmest Month BIO6 Min Temperature of Coldest Month BIO7 Temperature Annual Range (BIO5-BIO6) BIO8 Mean Temperature of Wettest Quarter BIO9 Mean Temperature of Driest Quarter BIO10 Mean Temperature of Warmest Quarter BIO11 Mean Temperature of Coldest Quarter BIO12 Annual Precipitation BIO13 Precipitation of Wettest Month BIO14 Precipitation of Driest Month BIO15 Precipitation Seasonality (Coefficient of Variation) BIO16 Precipitation of Wettest Quarter BIO17 Precipitation of Driest Quarter BIO18 Precipitation of Warmest Quarter BIO19 Precipitation of Coldest Quarter

We will add the following to the main text and will include the additional information listed in the above two tables in the supplemental materials;

"Using ARCGIS we clipped the output for each of these models to the study area defined in figure 1 and included them in our database. This includes layers for monthly average minimum temperature, monthly average maximum temperature, monthly total precipitation, and 19 Bioclimatic variables derived from WorldClim data (Hijmans et al., 2005 :Source: http://worldclim.org/bioclim )."

3. Is it the average of the multiple model outputs or just a single model output (which model) for 6 ka?

We illustrated our figure 7 with June Precipitation reconstructions using the Beijing Climate Center Climate System Model (BCC-CSM1.1).

See answer to #4 below for how we are adjusting this figure.

4. Although the wide range of a single climate variable (e.g. June precipitation in the text) at the study site is somewhat useful to illustrate the difficulty using a single site in assessing the impacts of the 4.2 event, the changing direction and rates are more important to assessing the influence of the event. So, the varied climate changes observed in the study area in different years might be more robust to illustrate this difficulty. In addition, the paleoclimatic proxy usually reflect the mean climate condition and the relative changes, so using the absolute value of June precipitation to illustrate the complex of using single site in studying the climate change is some unsuitable. Maybe using the seasonal or annual climate condition changes is more appropriate.

You are correct in that we used the single climate variable (June precipitation reconstructed from the Beijing Climate Center Climate System Model (BCC-CSM1.1)). Our choice was predicated on illustrating the difficulty in using a single climate variable. We also decided to use the BCC-CSM1.1 model since we felt that it might be tuned better for eastern Asia. Since monsoonal precipitation is important here we wanted to use a

variable that reflected water availability. Hence, our original choice of June precipitation for figure 7. However, and as you note, it might be more suitable to use an annual or seasonal climate condition. The bioclimatic reconstructions for 6 ka BP include; Annual Precipitation, Precipitation of Wettest Month, Precipitation Seasonality (Coefficient of Variation), Precipitation of Wettest Quarter, Precipitation of Driest Quarter, Precipitation of Warmest Quarter and Precipitation of Coldest Quarter.

We have chosen to use Annual Precipitation (Bio12) from the Beijing Climate Center Climate System Model in the revised version of our figure 7. This will also be noted in the caption for that figure.

5. In the 77 sites reported in 60 published papers, some sites might be reported several times, which may bias the evaluation inevitably. How to deal with these repeats in different publications should be considered in the geospatial analysis.

We agree that repeat references to the same site in publications are difficult to deal with. In some cases, there is a simple repeat from one publication to the next. This can be eliminated through careful evaluation of each paper included in the database. In other cases, new analyses are added as a given author continues to work a given site and it is valuable to have each new record as well as the new analysis and interpretation for that site included. We have attempted to minimize duplication in our review of articles included in the database by careful review of each paper. We acknowledge that there is no simple fix for this issue - a weighting function in later versions of the database may be advisable. We also note that duplicates can also be viewed as a positive feature of the database- indicating sites with well-established records that have been peer reviewed multiple times.

We have added the following text following line 8 on page 6 where we discuss database creation.

". . ..in the articles analyzed. Duplicate records reflecting multiple papers dealing with the same site were eliminated unless we believed that the given paper introduced a

new analysis/interpretation for a given site."

7. The 4.2 ka event is hardly to be extended to 3.0 ka and even later. A return to grass land condition between 2.8 and 1.5 ka BP at eastern Hunshandake can't be regarded as a different signal for 4.2 ka event. So, I suggest the authors should also double check the response of 4.2 ka event at other sites and place this event within a certain period, although the chronology uncertainty could be a factor broadening this period slightly.

We agree that chronology (dating) uncertainty presents another important issue and can be difficult to understand, let alone capture within a database. As well, a further difficulty is introduced with respect to the time-transgressive nature of the events that we are attempting to document. The 4.2 ka event appears in some areas to have multiple stages while in others only a single event is discernable. Additionally, and possibly just as important, is the underlying spatial diffusion and autocorrelated nature of the paleoclimatic variables that we and other researchers capture in our field analysis and measurement.

The temporal (dating and time transgressive) and spatial (diffusion) "fuzziness" of pale-oclimatic reconstructions in general makes records using different methods, or records from different areas, difficult to compare. We speculate that some of the uncertainty with respect to the 4.2 ka event results from this fuzziness. We believe that compilations of evidence for the 4.2 ka event, like the one that we present in this paper and database for northeast China, help to reduce this uncertainty.

To address these points, we have added the following text in our conclusions on page 10 line 18;

"….indicators analyzed and reported. We note three important issues that "broaden" any palaeoclimatic estimate and introduce uncertainty, namely, 1) Temporal Resolution- different temporal resolution between different dating techniques, 2) Localization versus Regionalization- the fact that some measures are "local" while others integrate "regional" conditions (for instance sediment in a small lake basin versus a lake that is the terminal sink of a large area respectively) and, 3) Lagged Response- the possibility of differential lagged responses for different measures of the same event. While much work remains . . . . . . ."

―――――――――――――――――――